# Predicting vaccine effectiveness for mpox

**Matthew T. Berry[1], Shanchita R. Khan ●[1], Timothy E. Schlub ●[1,2], Adriana Notaras[1], Mohana Kunasekaran[1], Andrew E. Grulich[1], C. Raina MacIntyre[1,3], Miles P. Davenport ●[1] ✉ & David S. Khoury ●[1] ✉**

The Modified Vaccinia Ankara vaccine developed by Bavarian Nordic (MVA-BN) was widely deployed to prevent mpox during the 2022 global outbreak. This vaccine was initially approved for mpox based on its reported immunogenicity (from phase I/II trials) and effectiveness in animal models, rather than evidence of clinical efficacy. However, no validated correlate of protection after vaccination has been identified. Here we performed a systematic search and meta-analysis of the available data to test whether vaccinia-binding ELISA endpoint titer is predictive of vaccine effectiveness against mpox. We observe a significant correlation between vaccine effectiveness and vaccinia-binding antibody titers, consistent with the existing assumption that antibody levels may be a correlate of protection. Combining this data with analysis of antibody kinetics after vaccination, we predict the durability of protection after vaccination and the impact of dose spacing. We find that delaying the second dose of MVA-BN vaccination will provide more durable protection and may be optimal in an outbreak with limited vaccine stock. Although further work is required to validate this correlate, this study provides a quantitative evidence-based approach for using antibody measurements to predict the effectiveness of mpox vaccination.

Mpox (formerly monkeypox) is a disease caused by the monkeypox virus (a zoonotic virus) that is endemic in West Africa with significant outbreaks occurring in 1980–1986 and 1997–1998[1]. Prior to 2017, these outbreaks were typically small and initiated by zoonotic transmission followed by self-terminating human-to-human chains of transmission[2]. However, since 2017, there has been a resurgence of mpox in Nigeria, Democratic Republic of the Congo (DRC) and other parts of Africa, attributed to waning immunity from smallpox vaccines and accumulation of cohorts that have never been vaccinated against smallpox[3]. In 2022, a global outbreak of mpox resulted in 91,000+ confirmed cases in 115 countries and established chains of human-human transmission leading to a renewed focus on vaccination as a preventative measure for mpox[4].

Although there is no mpox-specific vaccine, first generation smallpox vaccination was observed to protect individuals against mpox infection during the 1980–1986 mpox outbreak in the DRC

(then Zaire)[5–8], with an estimated vaccine effectiveness of approximately 85%[5], and this has also been observed in similar subsequent studies[9–11]. However, the live-replicating vaccinia vaccines (first and second-generation) have significant risks of serious vaccine adverse events[12], which led to the development of the third-generation Modified Vaccinia Ankara live-attenuated (replication deficient) vaccine (MVA-BN). Prior to the 2022 mpox outbreak, MVA-BN was approved by the FDA for use as a smallpox and mpox vaccine (two doses of $1 \times 10^8$ TCID via subcutaneous injection). Given the challenge of directly assessing the efficacy of this vaccine in an RCT, regulatory approval was based on demonstrated non-inferior immunogenicity profile and improved safety compared to the second-generation ACAM2000 vaccine[13]. In particular, comparing vaccinia neutralizing antibody titers induced by vaccination of MVA-BN and ACAM2000, it was deemed "reasonable to expect that this regimen of the vaccine is effective in smallpox vaccinia-naïve as well as in smallpox vaccine

[1]Kirby Institute, University of New South Wales, Sydney, NSW, Australia. [2]Sydney School of Public Health, Faculty of Medicine and Health, University of Sydney, Sydney, NSW, Australia. [3]College of Public Service and Community Solutions, and College of Health Solutions, Arizona State University, Tempe, AZ, USA. ✉e-mail: m.davenport@unsw.edu.au; dkhoury@kirby.unsw.edu.au

experienced individuals"[13]. This was supported by studies in nonhuman primates implicating antibodies directly in mediating protection against lethal mpox challenge[14].

Analysis of case data during the 2022 global outbreak indicates that the MVA-BN vaccine is effective for prevention of mpox[15–20], and affirms the decisions to use these vaccines during the outbreaks. However, important questions remain to be addressed. Firstly, how does MVA-BN effectiveness compare with the protection conferred by the live replicating smallpox vaccines, and how many doses are required? Further, is the protection from MVA-BN vaccination expected to be durable, and will further booster doses be required to confer durable protection against mpox and protect individuals in potential future outbreaks?

Here we address these questions by aggregating the available data on the effectiveness of different vaccinia-based vaccination regimens in protection against mpox. We compare protection from first generation smallpox vaccines with the protection conferred by one or two doses of the MVA-BN vaccine. Further, given the assumed role of antibodies in protection, we aggregate data on vaccinia-specific ELISA endpoint titers (here after referred to as vaccinia-binding titers) after MVA-BN vaccination to test for an association with protection. Finally, we analyze the kinetics of antibody decay over time to predict the duration of protection afforded by 1, 2 or 3 doses of vaccination. This work offers a data-driven approach to support public health decision making on mpox vaccination and boosting campaigns.

## Results

### Search results of vaccine effectiveness and immunogenicity studies

Our search identified 14 studies of vaccine effectiveness against mpox that met the inclusion criteria (Fig. S1). These studies included analysis of secondary contacts ($n = 5$), case-coverage studies ($n = 5$), a cohort study ($n = 1$), and case-control studies ($n = 3$) (Table S1). Of these studies, seven reported vaccine effectiveness from first generation smallpox vaccination against mpox (5 secondary contacts, 2 case-coverage), three determined effectiveness after one-dose of MVA-BN only (2 case-coverage, 1 cohort study), and four studies included protection from both one and two doses of MVA-BN (1 case-coverage, 3 case-control). One study by ref. 21 was excluded, because a more recent report by the same authors was identified that contained more data[17]. Further, another 2 studies[5,8] were excluded as they were all performed using similar secondary contact data from the Democratic Republic of Congo during the 1980−1986 outbreaks and thus we used only the study providing the most detailed disaggregation temporally and by age[7]. Information on the timing of cases and follow-up following vaccination was usually not reported (Table S1). For first generation vaccines the observation periods started after routine vaccination ceased in the area[9] and subsequent infections occurred long after vaccination. The case-control and case-coverage studies of MVA-BN monitored infections during a period of ongoing vaccination. Only individuals who were vaccinated more than 14 days ago, were considered vaccinated within these studies, however the time between vaccination and infection was not reported. In the single cohort study, participants had 21 weeks follow up time with infections in vaccinated individuals occurring at 3 weeks and 5 weeks post-vaccination[18].

Our systematic search for immunogenicity data yielded 43 clinical trials that reported on vaccinia-binding titers after MVA-BN vaccination (Fig. S2). We focused on vaccinia-binding titers because Zaeck et al have shown that endpoint antibody binding titers correlate well with neutralizing antibody titers against monkeypox virus, using samples from recently vaccinia-vaccinated individuals (reported $r = 0.82$, $p < 0.0001$)[22]. Moreover, endpoint vaccinia-binding titers are the most commonly reported measure of immunogenicity allowing comparison between multiple studies. A subset of 12 trials were identified that reported vaccinia-binding titers in healthy individuals[23–35], and one

further trial was published only on the clinicaltrials.gov database[36] (Table S2). These trials used similar methodologies to assess vaccinia-binding, allowing comparison of immunogenicity between trials. Two studies contained data on all three relevant groups (historic smallpox vaccination, 1 dose MVA-BN and 2 dose MVA-BN vaccination), and eight studies contained data for both the MVA-BN 1-dose and 2-dose groups.

### Vaccine effectiveness against mpox

Using the studies identified by our systematic search, we performed a meta-analysis to estimate an aggregate vaccine effectiveness (VE). In this analysis, we stratified data by vaccine type (i.e., historic first-generation, or recent 1-dose or 2-dose MVA-BN vaccination), and a hierarchical model structure was used to account for study heterogeneity (Fig. 1A and Table S3). Aggregating the available data, we used a Bayesian hierarchical model to obtain best-estimates of the effectiveness for historic first-generation vaccines (73.6%, CI:49.0−85.8%), one dose of MVA-BN (73.6%, CI:50.2−83.5%) and two doses of MVA-BN (81.8% CI:65.0−89.1%). We observed that one dose of MVA-BN provided similar effectiveness than historic vaccination, whilst two doses provided higher effectiveness, though the results were not significant (OR = 1.00, CI:0.46−2.44, OR = 0.68, CI: 0.31−1.69, respectively). Importantly, we had limited power to detect such a difference − and this is reflected by the large credible intervals. Despite the variation in reported VE across different studies (ranging from 35.8 to 86.4% for 1 dose and 66−89.5% for 2 doses, Fig. S3), we observed a significant benefit of two-dose vaccination over one-dose vaccination (OR = 0.69 (CI: 0.55-0.86)) − evident because four studies compared VE after 1 and 2 doses, and all four showed a trend for higher VE after 2 doses (Fig. 1B).

### Vaccinia-binding titers in vaccinated individuals

To investigate the immunogenicity of different vaccination strategies, we aggregated data on the geometric mean vaccinia-binding titers (GMT) reported 4 weeks after one dose of MVA-BN and 2 weeks after two doses of MVA-BN. This is because these were the most common times sampled after first and second doses of MVA-BN vaccination across all studies (Table S2). This also coincides with when the peak titers were observed after each subsequent dose[23–25]. For first generation vaccines, we use the baseline immunogenicity data (prior to receiving MVA-BN vaccine) for groups who had evidence of a previous smallpox vaccination. This is assumed to reflect the long-term vaccinia-binding titers maintained by individuals after receiving a first generation vaccine many years earlier. Our analysis aggregated data from studies that used slightly different ELISA assay protocols to estimate antibody titers (Table S2). We fit a Bayesian model with covariates for vaccine formulation and ELISA assay used, and a hierarchical structure to account for interstudy variability (see Methods, Table S4). Interestingly, we found no significant effect of the ELISA assay used to measure the GMT (Fig. S4a), but the formulation of the vaccine had a significant effect on the antibody titer (freeze dried formulation provided higher titers than liquid frozen, Fig. 2, $p < 0.001$) (Fig. S4b). Since the liquid frozen formulation was the formulation deployed in the effectiveness studies, we hereafter focus our analysis on this formulation. After accounting for all study and formulation differences, we found that the GMT induced by one dose of liquid frozen MVA-BN provides a higher antibody level than that observed in individuals who received a first-generation vaccine historically (GMT = 87.2 (CI: 66.9−115) vs 58.7 (CI: 41.4−82.2), fold difference 1.49 (CI:1.16−1.92)), and a second dose of liquid frozen MVA-BN provided a significant boost over a single dose (8.42-fold (CI: 7.79−9.10) increase, Fig. 2).

### Estimating a quantitative relationship between vaccinia-binding titers and vaccine effectiveness

Antibody titers are thought to be a surrogate of vaccine effectiveness for both smallpox and mpox infection[14,37], and were used to support

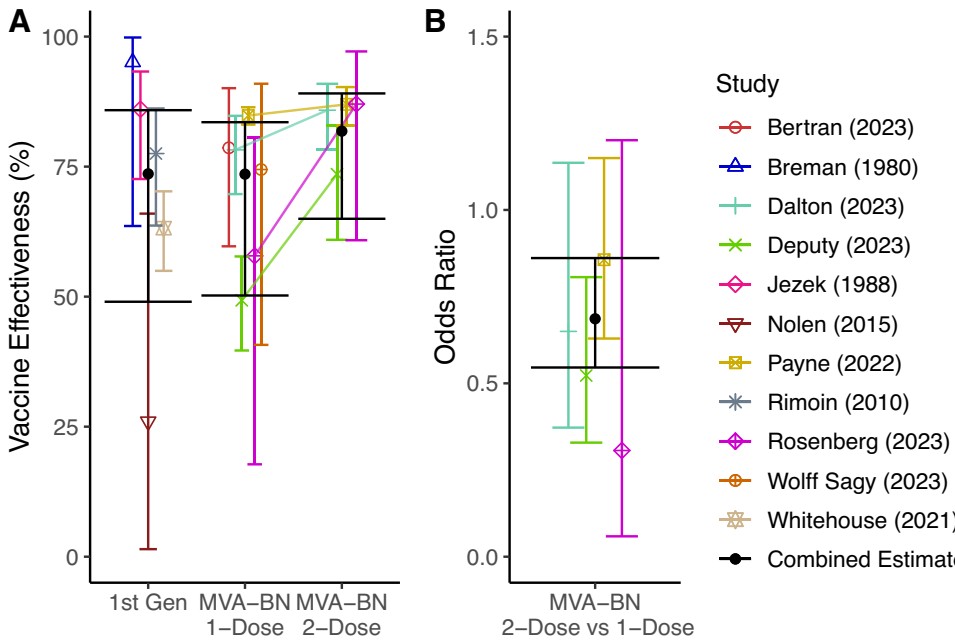

**Fig. 1 | The estimated effectiveness of different mpox vaccine regimens. A** The estimated effectiveness of first-generation vaccines administered years earlier (1st Gen, $n = 5$), as well as recent vaccination with a single dose of MVA-BN ($n = 6$), and two doses of MVA-BN ($n = 4$) are shown. The coloured points show the estimated effectiveness (median of the posterior distribution) for each regimen and study, the error bars indicate the corresponding 95% credible intervals. Where two regimens were compared in the same study, the effectiveness of the two regimens is joined by a line. The combined estimates (median of the posterior distribution) and 95% credible intervals for each regimen across all studies are shown in black. **B** The additional protection (odds ratio) provided by a two-dose regimen compared to a one-dose regimen ($n = 4$). The combined estimates (black) are the medians of the posterior distribution (circle) with the 95% credible intervals (error bars).

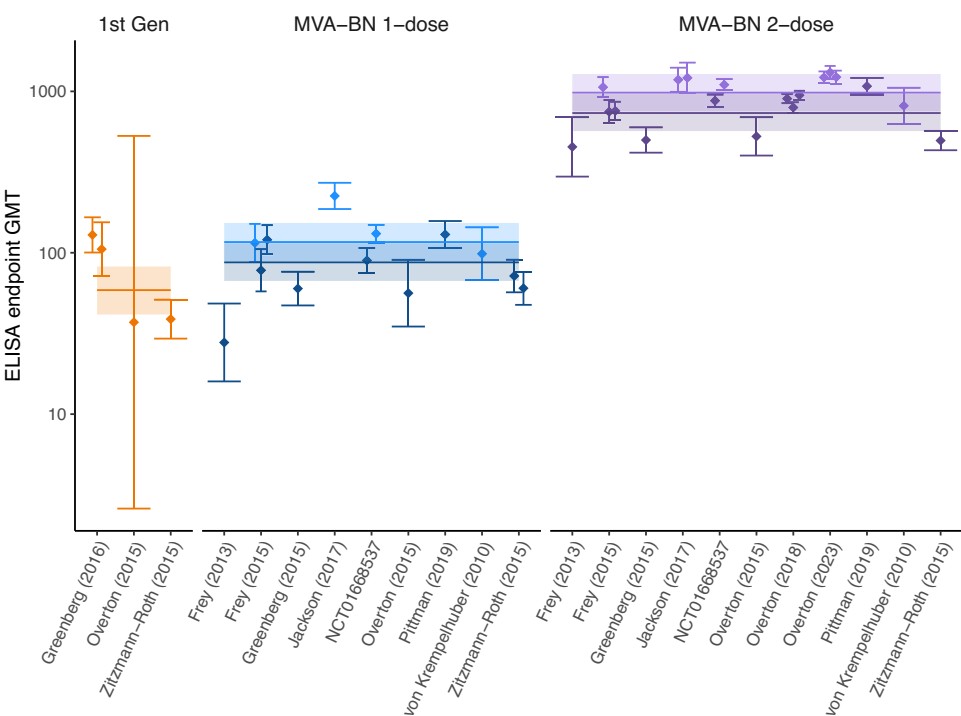

**Fig. 2 | Comparison of the reported geometric mean vaccinia-binding titers induced by vaccination with MVA-BN ($n = 12$) and historic first-generation vaccines ($n = 3$).** The GMTs from the freeze-dried MVA-BN formulation ($n = 5$, light-color) are higher than the liquid frozen formulation ($n = 8$, dark-color). That is, the ratio of titers in the freeze-dried and liquid frozen formulations (median of the posterior distribution) is 1.32-fold (95% credible interval:1.20−1.48). The points and error bars indicate the GMT reported in each study, along with 95% confidence bands extracted from each study respectively. Horizontal lines indicate the combined estimate (median of the posterior distribution) for each vaccination and formulation (shaded regions are the 95% credible intervals). The different colours represent the different vaccines with one and two doses of MVA-BN coloured differently.

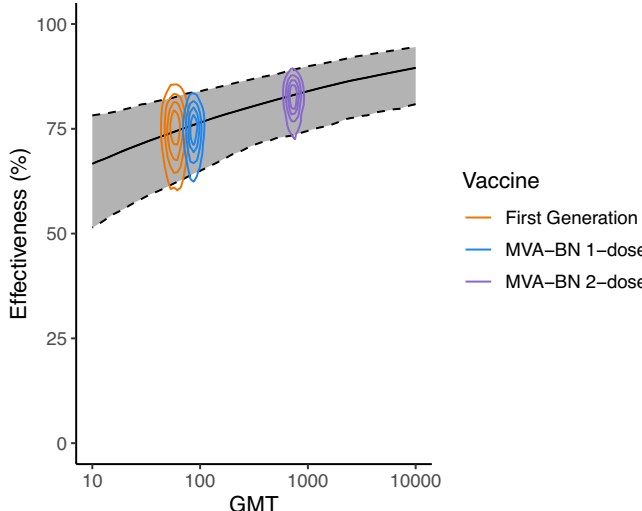

**Fig. 3 | Relationship between vaccine effectiveness and the vaccinia-binding GMT.** The contour lines, represent the 20%, 40%, 60% and 80% highest density regions of the joint-posterior distribution (i.e., smallest areas that contain x % of the posterior samples) for the different vaccines (indicated by colour). The association between antibody titers and effectiveness (solid black line) is fitted using all of the underlying data (accounting for the interstudy heterogeneity using a hierarchical model structure) (Table S5). The solid black line indicates the best estimate (median of posterior), and shaded region show the 95% credible intervals of the predicted effectiveness at different GMTs.

the regulatory approval of MVA-BN[13]. This decision was supported by animal studies showing an important role of antibodies in protection from mpox[14]. Thus, we sought to investigate the relationship between vaccinia-binding antibody titer and effectiveness by combining the available immunogenicity data and effectiveness data. No study found in our searches contained both effectiveness data and immunogenicity data from the same cohort. Therefore, we matched the immunogenicity data to the corresponding vaccine effectiveness data by vaccine regimen (Fig. S5). That is, one-dose MVA-BN vaccination ($n = 9$ immunogenicity, and $n = 6$ effectiveness studies, respectively), two-dose MVA-BN vaccination ($n = 11$, and $n = 4$) and historic first-generation smallpox vaccination ($n = 3$, and $n = 5$). The VE studies on MVA-BN were conducted during a period of ongoing vaccination and in most cases did not report the time between vaccination and infection (Table S1). Subsequently, we match this effectiveness data to the peak vaccinia-binding titers, which occur shortly after vaccination. First-generation vaccines were administered in the DRC prior to 1980[9] (when routine vaccination officially ceased), whilst the effectiveness studies range between 1970 and 2015. Subsequently in some studies vaccination occurred more than 35 years prior to infection. In order to account for these effectiveness studies being in individuals many years post-vaccination, we match this effectiveness data with the immunogenicity data for historically vaccinated cohorts (i.e., individuals vaccinated years earlier, and enrolled in an MVA-BN vaccine trial, but we use their baseline vaccinia-binding titers before they receive the MVA-BN vaccine).

Despite having very limited data to assess a correlation (i.e., only three vaccine groups across 14 effectiveness and 13 immunogenicity studies), fitting a logistic relationship between antibody titers and effectiveness in these groups (following the approach used in COVID-19[38,39]) (Fig. 3), we found evidence of a significant positive association between antibody titers and effectiveness (OR: 0.49 (CI 0.21−0.79) for each 10-fold change in vaccinia-binding, $p < 0.001$) (Table S5). This supports the use of vaccinia-binding titers as a correlate of vaccinia-based vaccine effectiveness. Further, this model provides a quantitative method to predict vaccine effectiveness (along with credible

intervals) associated with different antibody titers and waning immunity.

## Boosting and waning of antibody titers with MVA-BN vaccination

A major question regarding mpox control is how to optimize vaccine distribution and dosing intervals in the context of a potential future outbreak. Estimates of both the durability of protection and of the effects of the interval between first and second dose of MVA-BN would be informative in guiding policy for future responses. The immunogenicity studies we identified included a subset of studies that reported long-term follow up of antibody titers after MVA-BN vaccination (up to 24 months), as well as the effects of different timing of a second MVA-BN dose[24,25,27−36]. To explore the effect of dose timing on the peak and durability of antibody responses, we fitted a two-phase antibody decay model to the available vaccinia-titers over time for different vaccination regimens (Fig. 4A and Table S6). From this analysis we firstly noted, consistent with other vaccines[40], increased spacing between MVA-BN vaccine doses led to a higher peak antibody response (measured at 14 days post-last dose or 28 days post-initial dose, whichever is later) (Fig. 4B). For example, delaying the timing of the second dose from 7 days to 28 days led to a 4.2-fold (CI: 2.1−8.8) higher antibody titer. Delay from 28 days to 730 days led to a 3.2-fold (CI:2.6−3.8) higher titer. Interestingly, the peak antibody titer after a second dose of MVA-BN at 730 days was very similar to the peak titer of a third dose of MVA-BN at 730 days (after an initial 28 day-spaced two-dose regimen) (GMR 1.03, CI:0.82−1.31).

Regarding decay, we observed a fast initial decay and slow long-term decay of antibody titers (i.e., model comparison indicates that a two-phase decay model is superior to a single-phase decay, Table S7), and the estimated half-life of the fast-decaying and slow-decaying antibody titer was 20.7 (CI:18.2−24.0) days and 1721 (CI:971−6459) days, respectively. The estimated decay rates of the fast and slow-decaying antibodies were not different between the regimens (Table S7). However, the initial antibody titer and proportion of slow-decaying (long-lived) antibodies varied between groups (Fig. 4 and S6, discussed below). Interestingly, the proportion of long-lived antibodies increased in individuals with a 730-day spacing between first and second dose, when compared with the standard two-dose schedule (Fig. S6 and Table S8). Thus, when we consider the predicted antibody titers one year after boosting, delaying the second dose to 730 days provides a 14.1 (CI: 10.9−18.3) fold higher titer compared to standard boosting at 28 days (Fig. 4C and Table S8). Interestingly, whether vaccination at 730 days was given as a second booster, or as a third booster (after a second at 28 days), the durability of the response was similar (GMR 1 year after peak of 3 doses to 2 doses: 1.2 (CI:0.89−1.6)).

Of note, we estimate that antibody titers after two doses of MVA-BN vaccination (with 28-day interval) remain above or equal to the peak GMT of one dose for 81 days (CI:72−93) and above historically vaccinated cohorts for 102 days (CI: 74−173). By delaying the second dose to two years, antibody titers remain above the one-dose peak for 13.2 years (CI:7.5−48.3 years). Together these results suggest that a delayed booster or third dose provides higher and more durable antibody titers.

## Predicting vaccine effectiveness

Using the decay kinetics of antibody titers estimated above (Fig. 4A), and the logistic relationship between antibody titers and vaccine effectiveness (Fig. 3), we can predict the duration of vaccine protection under different vaccine schedules (Fig. 4D). Our analysis predicts vaccination with one dose of MVA-BN will have an effectiveness of 64.8% (CI:47.8−76.9) at two years post-vaccination (note that this VE estimate is an extrapolation below the range of data in Fig. 3). A two-dose regimen on a 4-week schedule is anticipated to still provide 71.8% (CI:58−80.8) effectiveness from mpox infection at two years.

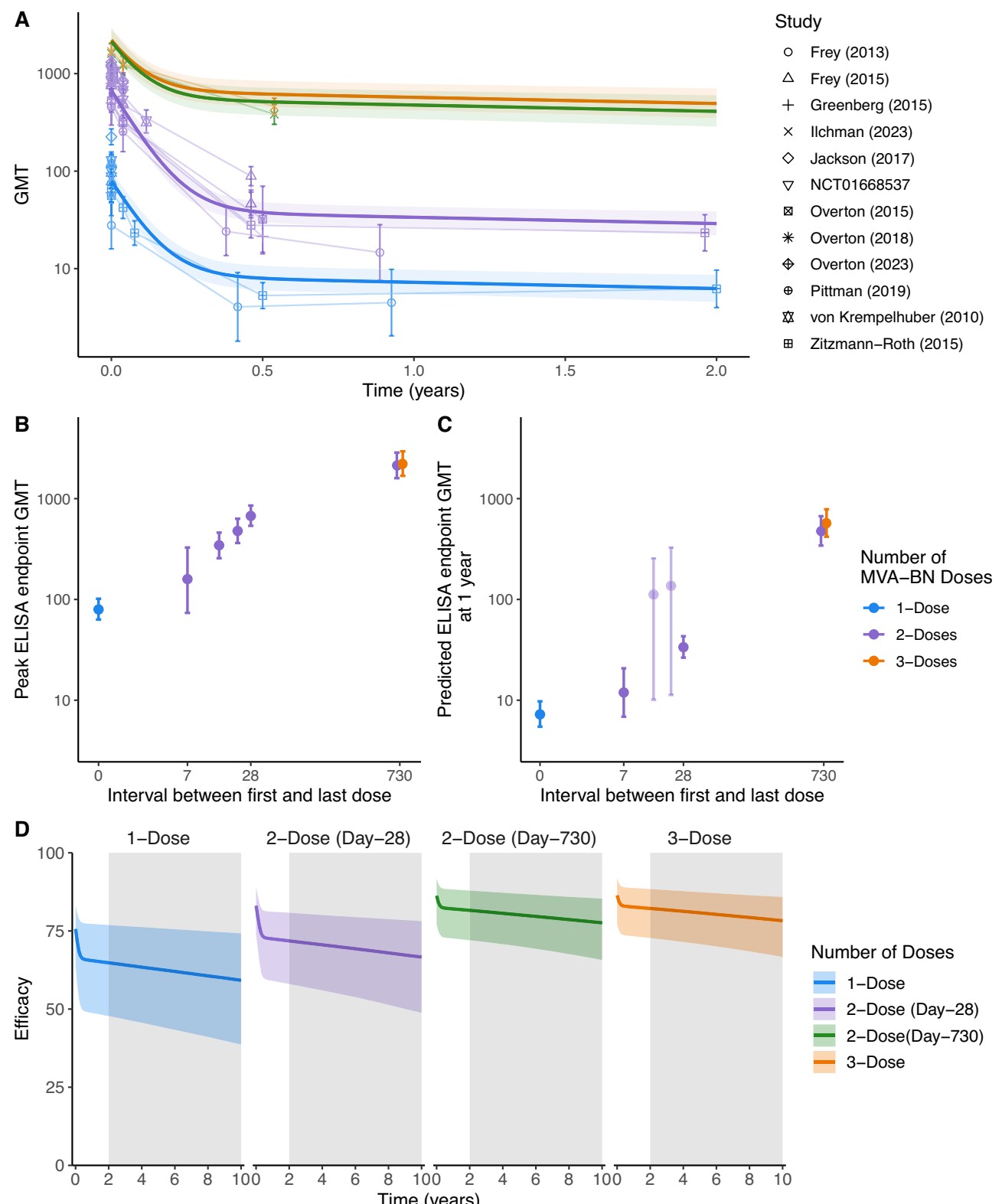

Extrapolating antibody decay beyond the available time series (shaded region in Fig. 4D), we also predict the level of long-term effectiveness from 1- and 2-dose MVA-BN vaccination (with 28-day spacing) after 10 years will be 59.2% (CI: 38.7−74.2) and 66.6% (CI:48.8−78.2), respectively. This is based on the conservative assumption that antibody decay continues at the same rate over 10 years (although studies of antibody decay after first generation vaccinia vaccination suggest the half-life may continue to slow to as long as 99 years[37]). If the second

vaccination is delayed to 730 days, then the predicted effectiveness at 10 years post-boost is 77.6% (CI:65.7−85.4). Together this analysis predicts long-term protection against mpox after one, and especially two doses of MVA-BN vaccination.

## Discussion

The ongoing spread of mpox in West and Central Africa, the 2022 global pandemic, and the associated changing epidemiology of

**Fig. 4 | Predicting the durability of protection. A** The decay in GMT in the different immunogenicity trials as fitted using a two-phase decay model across all immunogenicity studies ($n = 13$). The estimated GMT (median of the posterior distribution, solid lines) and 95% credible intervals (shaded) are shown over the two-year period for which immunogenicity data was measured in the one and two dose schedules (vaccine schedule indicated by colour). This is compared to the GMT (points) and standard deviation (error bars) reported in each study. **B** The effect of delayed dosing as estimated from fitting all the studies ($n = 13$) in (**A**). The median GMT (points) and 95% credible intervals (error bars) at the approximate peak (2 weeks after the final dose or 28 days after the first dose, whichever is later) is shown for different MVA-BN vaccination regimens. **C** The predicted GMT one-year post-vaccination (points) and 95% credible intervals (error bars), accounting for the early fast-decay and the late slow-decay of antibodies using the model fitted in (**A**) (Table S8). Predicted GMTs for regimens without a datapoint later than 5 months post-vaccination are shown with reduced opacity. **D** The predicted vaccine effectiveness (solid line) and 95% credible intervals (shaded area) over a 10-year period for the different vaccination schedules. The grey region highlights the prediction extrapolated beyond the available time course of immunogenicity data. The three-dose schedule involves vaccination on day 28 and two years after the initial dose (3rd dose delivered on day 730).

monkeypox virus highlight the importance of an improved understanding of mpox vaccination and immunity. Direct assessment of the efficacy of third-generation mpox vaccines has proved challenging due to the difficulties in performing large randomized controlled trials (RCTs). However, third-generation vaccinia-based vaccines were anticipated to be as effective against mpox as historical first-generation smallpox vaccines based on immunogenicity data[5,7]. Neutralizing antibody titers have been proposed as a correlate due to their role in protection from a lethal mpox challenge in animal models[14]. Subsequently, neutralizing antibodies and animal studies formed the basis for the approval of MVA-BN as a third-generation smallpox and mpox vaccine[13]. In this study, analysis of the vaccine effectiveness data indicates that single dose MVA-BN is non-inferior to historical vaccination with first-generation smallpox vaccines (OR = 1.00, (CI:0.46–2.44)). These findings support the choice to adopt MVA-BN vaccination for control of mpox given its comparable effectiveness to first-generation vaccines, and its beneficial safety profile[32]. Further, our meta-analysis finds two-dose MVA-BN vaccination to be more effective than one-dose (OR = 0.69 (CI: 0.55–0.86)).

A validated surrogate marker of mpox immunity would greatly assist in vaccine development and deployment, and in predicting the longevity of protection and necessity for boosting. In this work we aggregate the available data to study the relationship between vaccine immunogenicity and vaccine protection from mpox infection. We find a weak but significant association between antibody titers and vaccine effectiveness, albeit with very limited data. The significance of this association is predominantly influenced by the result that two doses of MVA-BN has both higher effectiveness and higher antibody titers than one dose. Caution is required before interpreting this result as a demonstration of antibody titers as a correlate of protection for vaccinia vaccines against mpox, since we have limited data available, a small range in observed effectiveness, and large interstudy variation. Our observation, however, supports the existing results from animal models that antibodies may be a correlate of protection against mpox[14].

If we assume that antibody titers are indeed a correlate, our model provides a means of predicting the long-term effectiveness given the available data (along with an estimation of confidence bands around this prediction). A major challenge during the 2022 global mpox outbreak was prioritizing the use of the small pool of existing MVA-BN vaccines. In particular, it was unclear whether improved overall outcomes would be achieved by, (1) maximizing the number of individuals who could receive a first dose, thus giving them some protection, or (2) focusing on maximizing the number of individuals who could receive the full two-dose regimen and ensure they had a sufficient immunological response for protection. Our analysis suggests that two doses of the vaccine provide only a slight increase in effectiveness compared to one dose (81.8% vs 73.6%). Therefore, in the context of a limited number of available doses, the increase in protection provided by a second dose (to recently vaccinated individuals) is less than the protection that could be obtained by giving a single dose to as many naïve individuals as possible. Assuming a population with equal risk, 1.79-times (CI:1.50–1.92) more cases could be averted by giving a single dose to twice as many individuals rather than a full two doses to a

smaller group. In addition, initially administering a single dose to the maximum number of people and delaying a second dose until there is increased availability of vaccine may provide additional benefits in terms of the durability of protection. For example, delaying a second dose until two years after the first dose is expected to provide a 3.2-fold higher peak titer and 14.1-fold higher titer at one year (compared to a second dose at 28 days) (Fig. 4A). Even though delaying a second dose produces a longer period of lower protection before boosting, protection from a single dose of MVA-BN is predicted to remain around 65% at two years (compared to 72% at two years for a two-dose regimen) (Fig. 4D). Further work is required to understand the optimal spacing of booster doses to maximize both short and long-term protection. However, these data predict that administering single doses initially allows the deployment of the vaccine more rapidly to more individuals during an emergency, and by delaying the second dose there is a potential advantage to the long-term durability of protection.

Our analysis includes a significant number of limitations. Firstly, the studies on vaccine effectiveness show a large amount of study heterogeneity (Table S1). For example, the estimated vaccine effectiveness after one dose of MVA-BN varies between 35.8% (CI:22.1–47.1)[16] to 86.4% (CI:83.3–89.0)[17]. This is perhaps not surprising, given the effectiveness data was obtained from observational studies with different study designs and potential confounders.

A major challenge in using non-randomized studies is appropriate matching of control groups (in case-control studies) and identification of the at-risk population (in case-coverage studies). Differences in matching cases to controls significantly affect the reported levels of protection[15,16,19]. In addition, large case-coverage studies attribute the reduction in case numbers to vaccination, but this may not control for confounders such as differences in behavior (which have been associated with a reduction in transmission in Italy prior to the commencement of vaccination[41]). Analysis of secondary contacts cannot always account for the number and significance of interactions with the contacts and thus can introduce unmeasured confounding. Further, in these studies conducted in the DRC, the status of individuals as vaccinated or unvaccinated was typically the result of the timing of a mass vaccination program that ceased in 1980[9] and so was heavily skewed by age. These confounders may contribute to the substantial heterogeneity in VE observed across studies (Fig. 1). We can partially account for unmeasured confounding by using a hierarchical model to account for inter-study variability, but systematic biases that result from unmeasured confounding are unable to be completely excluded. Further, the route and dose of MVA-BN administration varied over time and in different regions. For example, during the 2022 outbreak, following changes in FDA recommendations in the US, 45.9% of individuals received their first dose (and 85.7% received their second dose) via an intradermal (ID) injection of $2 \times 10^7$ TCID, instead of the per label recommendation of $1 \times 10^8$ TCID administered subcutaneously (SC)[42]. Our analysis of the effectiveness data could not test whether protection was impacted by the method of administration since the effectiveness studies did not disaggregate data by mode of administration (SC or ID). However, ref. 17., who tested for a difference in VE between individuals who received SC and those who received ID, reported no difference.

Another limitation is that the VE data and immunogenicity data came from independent cohorts and studies. Thus, there is no guarantee that the populations are well matched. Specifically, there is a significant mismatch in the demographics of the vaccine effectiveness studies and the immunogenicity trials. For example, the population considered in the US study for VE[17] considered only men (sex assigned at birth or gender identity) aged between 18 and 49. On the other hand, the clinical trials of antibody responses post vaccination were all tested on populations of both men and women and featured slightly different age ranges. Further, even though the majority (85.7%) of individuals in the US received their second dose of MVA-BN via intradermal administration[42], the majority of the available immunogenicity data was from individuals with SC administration (creating a potential mismatch in the effectiveness data and immunogenicity data). Fortunately, immunogenicity data suggests similar antibody titers between the two modes of administration[24]. An additional difference between immunogenicity and effectiveness studies was the timing of assessment of antibody titers in serum and effectiveness assessment. Whereas immunogenicity was assessed at 4 or 2 weeks after first or second vaccination respectively, the time from vaccination to infection is only reported in one effectiveness study (ref. 18., where 3 of 5 infections occur in week 3, and the other two infections occur in week 6). A mismatch also exists in comparing historic first-generation vaccination. The effectiveness data are from studies in the Democratic Republic of Congo in the 1980s to 2010s, who were vaccinated prior to 1980[9], whereas the immunogenicity data are from individuals in the United States (who in most cases were vaccinated >40 years earlier). As well as differences in the viral clades between these outbreaks[43,44], timing of previous vaccination in the historic vaccine effectiveness studies is not recorded and may not be well matched to the immunogenicity studies. Evidence of a very slow long-term decay of antibodies[37,45–47] suggests that time-since-vaccination may not be critical in comparing these groups many years after vaccination. However, it was not possible to match for age of vaccination, health status, or other demographic variables and their effects on immunogenicity and protection are unknown. Previous work has shown that vaccination in childhood confers longer protection than vaccination in adulthood[48]. Further investigation of the risk of breakthrough infection in historically smallpox vaccinated cohorts are required to confirm this assumption and improve our understanding of the duration of mpox immunity from vaccinia vaccines[49].

The evidence from animal studies showing a role for antibodies in protection from mpox[14], and the role of antibodies as a correlate of protection for smallpox[13], prompted us to consider antibodies as a correlate of protection in this study. However, despite finding an association between vaccinia-binding and protection, this is likely not an optimal correlate against mpox. In part this is because, even though vaccinia-binding titers are correlated with in vitro neutralizing antibody titers to mpox after primary vaccinia-vaccination[22], cross-recognition between vaccinia and monkeypox virus will likely be inconsistent when exposure histories vary (e.g., when an individual experiences a primary exposure to an mpox antigen rather than vaccinia[22]) or against different viral variants. Our ability to study neutralizing antibodies and other potential correlates of immunity was limited by the available data - with much less data on neutralizing antibodies than binding titers, and very limited reports and a lack of standardized assays for measuring cellular immunity. Further work is necessary to compare different measures of immunogenicity and define optimal correlates of protection for mpox.

Finally, it is not clear that a correlate of protection identified shortly after MVA-BN vaccination will continue to predict VE over time as immunity wanes. However, encouragingly, the data used in this analysis on the VE and antibody titers from first generation vaccinia-vaccination are all studying individuals long after vaccination and reveal similar titers and VE to a single dose of MVA-BN (Fig. 3), consistent with the possibility that vaccinia-binding continues to predict VE long after vaccination.

This study brings together the limited and heterogenous data available on immunogenicity and protection from mpox after MVA-BN vaccination. We report non-inferiority of MVA-BN against mpox compared to historic first-generation smallpox vaccination and define a candidate surrogate of VE based on vaccinia-binding titers. We then use that surrogate to predict the duration of VE. We predict that, since long-term vaccinia specific titers remain high, MVA-BN VE will remain >59% for up to 10 years, even after a single dose. This prediction of durable immune responses is consistent with reports of first- and second-generation vaccines against smallpox, where detectable immune responses and protection (particularly from severe infection) are thought to persist for over 20 years[50–52] (reviewed in ref. 48). Our approach allows the prediction of VE based on the existing available evidence, but also indicates the urgent need for further studies of MVA-BN immunogenicity and protection. The development of a standardized assay for mpox antibody binding or neutralization and a serological standard for comparison are important priorities. In the absence of this, comparison between studies by normalizing antibody levels to those induced by (for example) 1 dose MVA-BN vaccination (similar to methods used in COVID-19[38,39]) is possible but not ideal. In addition, public health plans for a potential future mpox outbreak need to be developed, ideally informed by the best available evidence for vaccine effectiveness.

## Methods

### Search strategy

We aimed to aggregate the available data that reported on both vaccine protection against mpox infection and vaccine immunogenicity in order to understand the relationship between immunogenicity and protection. For vaccine protection studies, we used a recently published systematic review[53] and extended this here. To obtain matching immunogenicity data, we performed a systematic search of ClinicalTrials.gov, EudraCT and ICTRP for all studies of MVA-BN immunogenicity. Screening of the results from each search were conducted independently by two individuals (MTB and SRK). Full details including search terms and inclusion criteria are provided in the supplementary material.

### Search strategy for vaccine effectiveness data

Our search strategy for identifying studies of VE was based upon the search strategy in the systematic review by ref. 53. This systematic review identified articles relating to mpox prior to 7th September 2020. We extend this by conducting a systematic search of PubMed (including MEDLINE) and Embase (Ovid) from the 7th September 2020 up to 10th July 2023 using the same search strategy as Bunge et al., with the following modifications:
- Added: "OR mpox[tiab]" to account for the recent name change,
- Added: "AND (Vaccine[tiab] OR Vaccination[tiab] OR Immunisation[tiab] OR Immunization[tiab])", since our search results are only targeted towards vaccine effectiveness (rather than all studies reporting on mpox)..

The full search terms for each database are provided in the supplementary materials.

Studies were included where they considered populations at-risk of mpox, where the intervention was vaccination with a vaccinia-based vaccine, the control or reference group was unvaccinated people (also at-risk of mpox), and where the outcome was mpox infection/incidence. For inclusion in our analysis, a study needed to include an estimate of VE, or report sufficient data such that VE could be estimated. This required data for:
- Cases of mpox in vaccinated and unvaccinated cohorts or populations along with an estimate for the population at risk for both

unvaccinated and vaccinated groups (for cohorts, case-coverage and secondary contact study types); or

– Vaccination status of individuals with cases of mpox, and a control group who were uninfected and for whom vaccination status was reported (for case-control studies).

Studies were excluded when vaccinia-vaccination was used as a post-exposure prophylaxis intervention rather than as a pre-exposure intervention.

## Search strategy for immunogenicity data

Since vaccinia-binding IgG antibody titers after MVA-BN vaccination have been shown to be highly correlated with neutralizing antibodies against vaccinia and monkeypox viruses[22], and because of the limited data on in vitro neutralization of monkeypox virus, we focused our search of immunogenicity data on vaccinia-binding titers. The immunogenicity data was obtained by searching through the ClinicalTrials.gov, EudraCT and ICTRP databases for clinical trials using the MVA-BN vaccine. We searched for intervention trials with "MVA" as the intervention with condition, "smallpox OR monkeypox OR Variola". We only considered studies that had been completed. For inclusion in our analysis, the intervention had to be MVA-BN (other MVA-based vaccines have been tested but are not used for immunization against mpox). Our analysis only considers healthy individuals. Therefore, at least one arm in the trial had to include a healthy population for the trial to be included in our analysis.

## Data extraction

For data on VE, we extracted the case/control incidence (number of events) data reported in each study disaggregated by timepoint, age or region where possible. For immunogenicity data, we contacted the sponsor by e-mail to request access to the de-identified individual-level data presented in the published work. This request was denied (27 March 2023). Therefore, we extracted summary data (Geometric Mean Titer (GMT) and confidence intervals (CI)) from tables (where available) or figures using WebPlotDigitiser[54]. Data extracted from tables was conducted by MTB and checked for accuracy by SRK. Where data was extracted from an image, two individuals (MTB, SRK) extracted the data independently with the geometric mean of the two extracted values used and we confirmed that discrepancies between extracted values were always less than 1%.

## Statistical analysis

All meta-analyses and meta-regression were performed using a hierarchical Bayesian data analysis framework in RStan[55,56] using the default HMC sampler. This included global estimates of mean effectiveness, and antibody titers after vaccination, estimates of the decay kinetics of antibody titers, and fitting the relationship between antibody titers and effectiveness. Hierarchical model structures were used to account for inter-study variability in all analyses, and unless otherwise stated all reported estimates are posterior medians, along with 95% credibility intervals (CI). To perform statistical tests of whether a parameter (or difference in parameters), g, was significantly different to zero we defined a $p$ value, $p = 2 \times \min(P(g>0), P(g<0))$, which was calculated from the posterior distributions of estimated parameters. Significance was defined as $p < 0.05$.

## Estimating vaccine effectiveness

In our meta-analysis of VE, we implement a modified version of the Bayesian binomial approach[27,28] to fit the raw event data from each study. The binomial model assumes individuals in a given population, N, have an unknown probability of infection, r. Therefore, the number of infections, n, follows a binomial distribution, $p(n|r) = \text{Bin}(r,N)$, with

posterior distribution,

$$p(r|n) = \frac{p(n|r)p(r)}{p(n)} \qquad (1)$$

We use an uninformative prior on the probability of infection, i.e., $p(r) \sim \text{Beta}(1,1)$. An equivalent model is used for the risk of infection after vaccination, $r_v$, with the vaccine effectiveness $E = 1 - OR$, where OR is the odds ratio (of the risk of infection between unvaccinated and vaccinated individuals). Unvaccinated individuals are used as the reference group for calculating vaccine effectiveness. When comparing relative effectiveness between vaccines we compare the odds ratio between the two vaccines (i.e., $OR_{1,2} = OR_1/OR_2$). Under the assumption of a rare disease ($r \ll 1$), the odds ratio, risk ratio and hazard ratio converge.

For all studies, we used the most temporally disaggregated data available (Table S1). This allows us to better account for changes in incidence and vaccine coverage over the outbreak. We also used data disaggregated by demographic factors, such as age, where available. Groups or timepoints in which no cases were reported in both control and vaccinated populations were excluded from the analysis. We assumed that the baseline risk of infection in unvaccinated individuals could differ for temporally or spatially disaggregated groups from the same study, but we assumed the vaccine effectiveness was constant. That is, for a given study, s, we had pairs of data $(n_{i,v,s}, N_{i,v,s})$, where, $n_{i,v,s}$ is the number of infections and $N_{i,v,s}$ the total individuals of the disaggregation group $i$, with vaccine $v$, in study $s$. We estimate a different baseline risk in unvaccinated individuals ($v=0$), $r_{i,0,s}$, for disaggregation group $i$, and a fixed vaccine effectiveness $E_v$. The risk of infection after vaccination ($v \neq 0$) is then given by,

$$\text{logit}(r_{i,v,s}) = \text{logit}(r_{i,0,s}) + \log(1 - E_v) + S_s \qquad (2)$$

where $S_s$ is the random effect on the risk reduction from vaccination (on the logit scale) in each study. These random effects are defined by the hierarchical structure $S_s \sim N(0,\sigma)$ (further detail in supplementary methods), where $\sigma$ is to be estimated. The posterior distribution is defined using the likelihood function,

$$n_{i,v,s} \sim \text{Bin}(r_{i,v,s}, N_{i,v,s}) \qquad (3)$$

and with the priors,

$$
\begin{aligned}
r_{i,0,s} &\sim \text{Beta}(1,1), \\
R_v &\sim N(0,10), \\
\sigma &\sim \text{Half} - \text{Cauchy}(0.25).
\end{aligned}
\qquad (4)
$$

For the case-control studies, the at-risk population is not used. However, vaccine effectiveness can be estimated using the odds ratio comparing the odds of vaccination in the infection and control groups[57]. Subsequently we can use the same model (Eqs. 2 and 3) to fit the case-control studies (detailed in Supplementary methods).

## Estimating immunogenicity

Our goal here is to estimate the mean vaccinia-binding titer induced by vaccination in each group, as well as the spread (standard deviation) of those titers between individuals (since this is important when relating antibody titers to protection[38]). That is, we aim to estimate the mean, $\mu_v$, and standard deviation, $\sigma_v$, of the distribution of antibody titers induced in a population after vaccination for a given vaccine regimen, $v$. Since we do not have individual level data from each study, we are limited to using only three pieces of data for each group, from each study, to estimate these quantities. Specifically, the (log) GMT

reported after vaccination, $\bar{y}_{s,vf,E}$, the number of individuals in these groups, $n_{s,vf,E}$, and the standard deviation of the (log) titers in these groups, $\overline{sd}_{s,vf,E}$ (derived from the reported confidence intervals, Supplementary Methods) – in each study, $s$, and for each vaccine group, $v$ (accounting for different assays, $E$, and vaccine formulations, $f$). We assume that individuals' (log) vaccinia-binding titers after vaccination are normally distributed, i.e.,

$$y_{i,s,vf,E} \sim N\left(\mu_v + \mu_s + \mu_f + \mu_E, \sigma_v\right), \tag{5}$$

where the mean and standard deviation of titers after vaccination with vaccine $v$ are $\mu_v$ and $\sigma_v$ (respectively), and assuming this mean is influenced by: (i) random study effects, $\mu_s$ (from a hierarchical model structure, where $\mu_S \sim N(0, \sigma_s)$, and $\sigma_s$, is also a parameter that is inferred), (ii) a fixed effect, $\mu_f$, to account for potential differences in antibody titers induced by the two different vaccine formulations used for the MVA-BN vaccine (freeze dried and liquid frozen), and (iii) a fixed effect, $\mu_E$, to account for the effect of the different ELISA assays (but the latter was not a significant covariate, and removed from further analysis). We use the above model to estimate the distribution of the participant level vaccinia-binding titers, using the sample mean, $\bar{y}_{s,vf,E}$, and sample standard deviation, $\overline{sd}_{s,vf,E}$ from each group/study as sufficient statistics to estimate this distribution (i.e., we require no other information to estimate the participant level distribution of vaccinia-binding titers other than the available sample means and standard deviations - derivation in supplementary material, including likelihood function). We impose the below weakly informative priors on our parameters,

$$\begin{aligned} \mu_d &\sim N(0,10) \\ \log(\sigma_d) &\sim N(0,10) \\ \mu_E &\sim N(0,10) \\ \mu_f &\sim N(0,10) \\ \sigma_s &\sim \text{Half} - \text{Cauchy}(0,1). \end{aligned} \tag{6}$$

When antibody titers were below the limit of detection, this was handled in different ways across the different studies. Some studies had assigned values of 1 to the titers below the detection limit before calculating the GMT, whilst other studies assigned those values as half the limit of detection (assign 25 to the titer given the limit of detection (LOD) is 50). Subsequently, the reported GMTs use different methods for the calculation. For consistency in our analysis, we adjusted the GMTs from the latter studies (those who assign the values below LOD as 25) to reflect the former approach (assign values below the LOD as 1). Using the reported number of seropositive samples (those above the limit of detection), we can reassign the titers below the LOD from 25 and set those values as 1 with the resulting adjusted GMTs now being consistently calculated across studies.

**Predicting vaccine effectiveness over time**

We fit a model of biphasic exponential decay to the vaccinia-binding antibody titers. This model has two compartments, long ($x_l$) and short ($x_s$) lived antibody titers, which each are assumed to decay with rates $\delta_l$ and $\delta_s$, respectively. Thus, the total antibody titer at time, t, is given by $x(t) = x_l(t) + x_s(t)$, where,

$$\begin{aligned} x_s(t) &= x_0 f e^{-\delta_s t}, \\ x_l(t) &= x_0 (1-f) e^{-\delta_l t}, \end{aligned} \tag{7}$$

and where $x_0$ is the antibody titer at $t = 0$ (which is defined as a maximum of 14 days after final dose or 28 days after the first dose), and $f$ is the fraction of the initial antibody titer that is short-lived. After model comparison (Table S7), we found that the decay rates $\delta_l$ and $\delta_s$ are not significantly different between dosing regimens. However, we assume the initial antibody titer ($x_0$) and the fraction short-lived, $f$, differs for

each dosing regimen and is also different for historically vaccinated groups. The log (base 10) of the model and data were fitted in RStan as described above in the "Estimating immunogenicity" section. We impose the weakly informative priors on the decay model parameters.

$$\begin{aligned} \delta_s, \delta_l &\sim N(0,1) \\ f &\sim U(0,1) \end{aligned} \tag{8}$$

**Fitting the relationship between vaccinia binding antibody titers and protection from mpox**

To analyse the relation between antibody titers and protection we applied a model we have previously used to identify a correlate of protection for COVID-19[38]. The model assumes that there is a logistic relationship between the protection, $P$, experienced by a group of individuals with a given vaccinia-binding antibody titer, $x$, given by,

$$P(x) = \frac{1}{1 + e^{-k(x-x_{50})}}, \tag{9}$$

where, $x_{50}$ is the 50% protective titer and $k$ describes the steepness of the relation between the antibody titer and the protection.

We re-parameterize this function with the substitution, $A = -k(2.5 - x_{50})$, to provide better numerical stability during model fitting, in cases where the two parameters trade-off, i.e.,

$$P(x) = \frac{1}{1 + e^{-k(x-2.5)-A}} \tag{10}$$

For a given vaccinated population, we must consider the observed population distribution of vaccinia-specific antibody titers (with mean $\mu$ and standard deviation $\sigma$). The average protection over the distribution of titers is given by,

$$VE = \int_{-\infty}^{\infty} P(x) N(x|\mu,\sigma) dx, \tag{11}$$

where $N(x|\mu,\sigma)$ denotes the probability density function of the normal distribution. The data on antibody titers and effectiveness are fitted simultaneously to estimate all model parameters along with associated credible intervals (Table S5). The priors used and hierarchical structure (to account for inter-study variability) are also provided in Table S9.

**Reporting summary**

Further information on research design is available in the Nature Portfolio Reporting Summary linked to this article.

## Data availability

All extracted data used in this manuscript is available in the "Data" folder of the GitHub repository, https://github.com/iap-sydney/Mpox_ELISA_Effectiveness_correlates.

## Code availability

All code used in the analysis is publicly available on GitHub, https://github.com/iap-sydney/Mpox_ELISA_Effectiveness_correlates.

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

## Acknowledgements

This work is supported by an NHMRC program grant GNT1149990 (to M.P.D.), Investigator grants (GNT1173931 to A.E.G., GNT2016907 to C.R.M. and GNT1173027 to M.P.D.) and an NHMRC Centre for Research Excellence BREATHE (GNT2006595 to C.R.M.). DSK is supported by a University of New South Wales fellowship. The funders had no role in the design and conduct of the study; collection, management, analysis, and interpretation of the data; preparation, review, or approval of the manuscript; and decision to submit the manuscript for publication.

## Author contributions

M.T.B., C.R.M., M.P.D., and D.S.K. contributed to the conceptual development of the project. M.T.B., S.R.K., A.N., M.K., C.R.M., and D.S.K. contributed to data identification, extraction, and curation. M.T.B., T.E.S., M.P.D., and D.S.K. contributed to data synthesis and analysis. M.T.B., A.E.G., C.R.M., M.P.D., and D.S.K. contributed to interpretation and description of results. All authors contributed to manuscript preparation and editing and approved the final submission.

## Competing interests

C.R.M. is on the WHO SAGE Working Group on Smallpox and Monkeypox. The authors have no other competing interests to declare.

## Ethical approval

The analysis of publicly available de-identified clinical trial data, as performed in this study, was approved under the UNSW Sydney Human Research Ethics Committee (approval HC200242).
