## [Peer Review File · Nature Communications]

REVIEWER COMMENTS

Reviewer #1 (Remarks to the Author):

The authors used very few data points and very complicated model to build a model between vaccine effectiveness in Mpox and antibody. The message is new but their confidence intervals are wide. I have doubts about its contributions to practice, but it is by far the best evidence.

Methodology

1. Search Strategy:

a. Since this study relies on a systematic search of published data, I am curious about the databases that were examined. Typically, a minimum of two databases, with one from the US and one from Europe, are recommended. It appears that efficacy data were exclusively gathered from ClinicalTrials.gov, potentially omitting some randomized controlled trials (RCTs) that were not registered on this platform.

b. Additionally, seeking the assistance of an information specialist with expertise in medical literature could enhance the study's comprehensiveness. Specifically, the vaccine search block appears to be incomplete.

2. Selection criteria:

a. I don't understand the selection criteria for efficacy RCTs (lines 603-604).

b. As this study correlates efficacy with antibody titers, how did the author assess the comparability of these studies except for vaccine regimen (for example, age, gender, comorbidity)? If there are multiple efficacy studies matching one immunogenicity study or vice versa, how did the author determine which studies to include?

3. Data extraction: Was the entire process of study screening and data abstraction conducted with double-checks? It has been reported that 17% of data cannot be reproduced in meta-analyses [DOI: 10.1136/bmj-2021-069155]. Therefore, ensuring the accuracy of extracted data is crucial.

4. Statistical analysis:

We are not familiar with hierarchical Bayesian. This part need further check from statisticians.

But why the authors not use the raw reported antibody? Are they still comparable after the transformation?

Results

1. Flowcharts (Figure S1):

a. 67 studies excluded via automation tools. What tool was used? What are the exclusion criteria for that tool, and has any study validated the accuracy of this tool?

b. Please provide a detailed definition for each criterion. For instance, what is considered a wrong study design, and how is a wrong population defined?

2. Lines 116-120: Was no vaccination the reference group when estimating effectiveness for all groups? What is the average follow-up duration for each vaccine? Due to waning of efficacy, it is important to describe the duration when comparing effectiveness across different vaccines.

3. Lines 145-146: Why did the author choose to use GMT 4 weeks after 1 dose MVA-BN and GMT 2 weeks after 2 dose MVA-BN? When was the antibody titer for 1st Gen detected?

4. Figure 4: Most points were between 0 to 0.5 years, very few points were in 1 to 2 years. Is this model reliable? Why is confidence interval so narrow?

Discussion

1. Lines 322-333: The author recommended a longer interval for booster vaccination based on the humoral immunity evidence presented in this study. However, considering the significance of cellular immunity in formulating vaccination strategies, could the author provide any supporting evidence for cellular immunity?

2. Lines 361-391: The author mentioned significant differences among certain paired studies involving efficacy and immunity. Could the author include a table delineating the characteristics of these paired studies for readers to easily comprehend the distinctions? Additionally, has the author attempted to select studies that are seemingly comparable for sensitivity analysis to evaluate the robustness of the results?

Reviewer #2 (Remarks to the Author):

This paper by Berry et al uses a similar technique used by the same group for COVID-19 vaccines to link vaccinia binding antibody toters with vaccine effectiveness. The results are that antibodies

correlate with efficacy according to a logarithmic - linear distribution and that vaccine efficacy can be projected to be reasonably high after 2 doses for many years based on antibody decay kinetics.

The work is of high significance to the field because vaccination strategies to protect against subsequent MPox outbreaks are needed and optimization of protection amongst high risk groups is required.

The work represents a quandary of sorts. The analysis contains no major flaws and probably approximates the best analysis that can be done with available data to optimize Mpox vaccine allocation now and in future outbreaks. That said, the available data is much less useful than the data this group used to conclude that neutralizing antibodies are a good surrogate of protection against SARS-CoV-2. The authors do a beautiful job of outlining these limitations in the discussion which include the fact that the analysis only contains 3 types of vaccine studies (which is very low to form a meaningful correlation as in Fig 3), that neutralizing Abs are often not measured in most of these studies, that the studies include a mix of case-control & cohorts studies in different populations which are not well matched, that vaccine formulations vary over time, that behavioral confounders likely vary between study populations, and that other potentially important Mpox immune responses go unmeasured, For these reasons, the study does not provide strong enough evidence to establish antibodies as a formal correlate and the authors acknowledge this.

Overall, the conclusions are only somewhat supported by the analysis despite an adequate and thorough methodological approach. Despite the level of evidence being only low-moderate, the results are probably actionable because this is the strongest type of analysis available for the problem to my knowledge, and the problem is rather urgent.

I have some minor suggestions:

- 1) Show the strength of binding / neutralizing antibody correlation in past studies and describe in a bit more detail how these studies were done.
- 2) When describing VE, state over what time interval. Was this interval even the same in all listed studies?
- 3) Figure 1B is too squished and would be easier to read with a wider x-axis.

4) The horizontal lines in Fig 2 are hard to see. I suggest reformatting

5) Fig 3: "The contours represent the lines of equal probability of the normalized joint..... etc..." I do not know what this means. This should be stated with less jargon.

6) For bi-phasic decay, how was the timing of the inflection point selected?

7) I feel as if the optimization of timing of dose #2 is important and should be mentioned in the abstract

Reviewer #3 (Remarks to the Author):

The authors performed a systematic search and meta-analysis of immunological and observational studies for Mpox vaccines, to establish the correlation of protection (CoP) between an immune marker (vaccinia-binding ELISA endpoint titer) and vaccine effectiveness. The authors then estimated the antibody kinetics using aggregated data collected two years after the vaccinations, and predicting the long-term vaccine effectiveness against Mpox by extrapolating the antibody kinetics to 10 years with the previously estimated CoP. The study largely inherited methods used for several COVID-19 studies published by these authors.

My main concern is that one of the main claims of this study was supported by extrapolating the established CoP of vaccinia-binding ELISA endpoint titer and vaccine effectiveness from 2-year data to a longer time span up to 10 years. Although the authors used a separate antibody kinetic model to predict the antibody titer level at ten years, they used a constant CoP that was estimated using data collected relatively shorter after vaccination. The key question will be whether or not the CoP identified by the authors will be a constant over time. If not, then the extrapolated vaccine effectiveness may be biased because the CoP was completely different from the early after vaccination. Such issue has been seen in COVID-19 where ELISA and even neutralising antibody derived from inactivated vaccines against Omicron were barely detectable, but actually the vaccine effectiveness against fatal outcomes were still >90%. Therefore, the current study over simplifies this question with extrapolations using only a constant CoP, which will weaken their findings and claims.

As the authors included the long-term vaccine effectiveness predictions as a part of the study, it is important to clarify at what time the antibody and vaccine effectiveness were measured throughout

the manuscript. Current, these information were mostly unclear when reading the first four sections of results. For example, when the authors describing estimating the correlation between antibody and vaccine effectiveness for the same vaccine type/dose, it is unclear whether the time points were also matched for these estimates.

Figure 1 - Suggest the authors to demonstrate the two panels as forest plots, where both individual and pooled estimates for subgroups can be clearly demonstrate. Also, it would be helpful to also report the I statistics for each subgroup to assess the heterogeneity of individual estimates within each subgroups.

Vaccine effectiveness were all extracted from observational studies, which unavoidably faced various types of confounders. Especially, there were several studies were performed in close contacts. Therefore, it is important to address this limitation and discuss the potential impacts of measurements from these study designs.

REVIEWER COMMENTS

Reviewer #1 (Remarks to the Author):

The authors used very few data points and very complicated model to build a model between vaccine effectiveness in Mpox and antibody. The message is new but their confidence intervals are wide. I have doubts about its contributions to practice, but it is by far the best evidence.

We thank the reviewer for their overall assessment of this work as “by far the best evidence”, albeit with very apparent limitations in the available data, which has been used in our analysis. Given that regulators are already using these same data to make policy decisions, our work aims to provide the best available integration of these data and quantify the uncertainties (which are large) – and so appropriately inform regulatory decisions. We appreciate the reviewer’s comments below and have sought to, wherever possible, increase the rigour of the systematic search and quality of data extraction.

Methodology

1. Search Strategy:

a. Since this study relies on a systematic search of published data, I am curious about the databases that were examined. Typically, a minimum of two databases, with one from the US and one from Europe, are recommended. It appears that efficacy data were exclusively gathered from ClinicalTrials.gov, potentially omitting some randomized controlled trials (RCTs) that were not registered on this platform.

We thank the reviewer for their comment and apologise for the lack of clarity. We have now detailed the full search strategy used, and the databases that have been searched in the methods and supplementary methods. In our original submission, the effectiveness data was obtained from an existing systematic review by Bunge et al., (reference 53), and an additional search of the PubMed database (including the MEDLINE database) from 7th September 2020 to 10th July 2023 to capture more recent studies. In our extended search, we used the same search strategy published by Bunge et al. (with minor alterations to include updated name changes and a search block to focus on vaccination). In our revision we have now expanded our systematic search to include the Embase database as well as PubMed, since this was the other database used by Bunge et al. (full details in supplement, which capture an additional 116 studies but no additional eligible studies) to make our extended search between 7th September 2020 to 10th July 2023 include two databases, and fully consistent with the search performed by Bunge et al. This search of the Embase database captured 134 studies that were screened (by two authors), but no more additional eligible studies were identified.

The reviewer is correct, in our original submission the immunogenicity data reported in the submitted manuscript was obtained exclusively from clinicaltrials.org. We have now extended the search to include the European (EudraCT) and WHO (ICTRP) clinical trials registries. Searches of the new databases identified 8 and 38 additional results, respectively (with 33 duplicates). However, after screening none of these additional studies were found to be eligible for inclusion.

The methods and supplementary methods have been updated to describe the search strategy and databases searched in more detail:

Line 430: Our search strategy for identifying studies of VE was based upon the search strategy in the systematic review by Bunge et al.⁵⁶. This systematic review identified articles relating to mpox prior to 7th September 2020. We extend this by conducting a systematic search of PubMed (including MEDLINE) and Embase (Ovid) from the 7th September 2020 up to 10th July 2023 using the same search strategy as Bunge et al., with the following modifications:

- Added: "OR mpox[tiab]" to account for the recent name change,
- Added: "AND (Vaccine[tiab] OR Vaccination[tiab] OR Immunisation[tiab] OR Immunization[tiab])", since our search results are only targeted towards vaccine effectiveness (rather than all studies reporting on mpox).

The full search terms for each database are provided in the supplementary materials.

And

Line 458: The immunogenicity data was obtained by searching through the ClinicalTrials.gov, EudraCT and ICTRP databases for clinical trials using the MVA-BN vaccine. We searched for intervention trials with 'MVA' as the intervention with condition, 'smallpox OR monkeypox OR Variola'. We only considered studies that had been completed. For inclusion in our analysis, the intervention had to be MVA-BN (other MVA-based vaccines have been tested but are not used for immunization against mpox).

b. Additionally, seeking the assistance of an information specialist with expertise in medical literature could enhance the study's comprehensiveness. Specifically, the vaccine search block appears to be incomplete.

We apologise for the lack of clarity on these search terms. Since our vaccine effectiveness data came from an existing systematic review (Bunge et al.), which we extended to include more recent studies, we had previously only cited this systematic review without reporting the full search strategy and terms used in our work. We now provide the full set of databases searched and the full strategy used to search these databases for data on vaccine effectiveness. In order to ensure our vaccine search block is more complete, we have also added "immunisation[tiab] OR immunization[tiab]" into the search terms which captured 15 additional results but no additional eligible studies.

Supplement Methods: Our systematic search for vaccine effectiveness studies extended the search from Bunge et al.²⁸ by adding the search terms "AND (Vaccination[tiab] OR Vaccine[tiab] OR Immunisation[tiab] OR Immunization[tiab])" and "OR Mpox[tiab]". These additional terms limited the scope to papers that considered vaccination usage and updated the search for the change in name. We conduct this search from the date, 7th September 2020 to the 10th July 2023. Specifically, we searched the PubMed (including the MEDLINE database) and Embase (Ovid) databases, with the search terms:

PubMed:

(Monkeypox[MeSH] OR "Monkeypox virus"[MeSH] OR monkeypox[tiab] OR "monkey pox"[tiab] OR "varirole du singe"[tiab] OR "varirole simienne"[tiab] OR Mpox[tiab])AND (Vaccination[tiab] OR Vaccine[tiab] OR immunisation[tiab] OR immunization[tiab]) ;

Embase (Ovid):

('monkeypox'/exp OR 'monkeypox virus'/exp OR monkeypox:ti,ab OR "monkeypox":ti,ab OR mpox:ti,ab) AND (Vaccinatio:ti,ab'OR Vaccine:ti,ab OR Immunisation:ti,ab)

2. Selection criteria:

a. I don't understand the selection criteria for efficacy RCTs (lines 603-604).

We apologise for the confusion. The aim was to include all studies where Vaccine Effectiveness (VE) could be calculated. This requires certain data be presented within the studies and the required data depends on the type of study. We have now made the inclusion criteria for vaccine effectiveness studies:

Line 444: For inclusion in our analysis, a study needed to include an estimate of VE, or report sufficient data such that VE could be estimated. This required data for:

- *Cases of mpox in vaccinated and unvaccinated cohorts or populations along with an estimate for the population at risk for both unvaccinated and vaccinated groups (for cohorts, case-coverage and secondary contact study types); or*
- *Vaccination status of individuals with cases of mpox, and a control group who were uninfected and for whom vaccination status was reported (for case-control studies).*

And a detailed explanation is provided in the supplementary methods:

Supplementary Methods: In order to calculate vaccine effectiveness from a given study, and thus include the study in our analysis, we required the following data:

- *In the case of studies based on population surveillance data (case-coverage), the incidence of infection must be provided in both vaccinated and unvaccinated at-risk groups.*
- *In the case of secondary-contact studies, the total number of secondary-contacts/exposed individuals (disaggregated by vaccine status) must be provided, along with the number of secondary-cases in vaccinated and unvaccinated individuals.*
- *In the case of cohort studies, the cohort size of both vaccinated and unvaccinated groups must be provided along with the number of infections in each group.*
- *For case-control studies we require a break-down of infection data by vaccination status along with an uninfected control group split by vaccination status as a comparator.*

b. As this study correlates efficacy with antibody titers, how did the author assess the comparability of these studies except for vaccine regimen (for example, age, gender, comorbidity)? If there are multiple efficacy studies matching one immunogenicity study or vice versa, how did the author determine which studies to include?

We thank the reviewer for their comment highlighting the lack of clarity. It should be noted that multiple vaccine effectiveness studies and immunogenicity studies have been linked to

estimate each point in the correlation analysis (i.e. many-to-many matching), rather than a pairing of individual studies (i.e. not one-to-one). This is now made clear in a sup. figure showing which studies contribute to each vaccine regimen:

It was not possible to pair studies based on explicit matching of population demographics, as limited information was provided to achieve this. There were some instances where the immunogenicity and effectiveness studies are known to differ demographically – for example by sex. The limitations of the matching of population demographics between immunogenicity and vaccine effectiveness studies are detailed in the discussion:

Line 347: Another limitation is that the VE data and immunogenicity data came from independent cohorts and studies. Thus, there is no guarantee that the populations are well matched. Specifically, there is a significant mismatch in the demographics of the vaccine effectiveness studies and the immunogenicity trials. For example, the population considered in the US study for VE¹⁷ considered only men (sex assigned at birth or gender identity) aged between 18-49. On the other hand, the clinical trials of antibody responses post vaccination were all tested on populations of both men and women and featured slightly different age ranges. Further, even though the majority (85.7%) of individuals in the US received their second dose of MVA-BN via

intradermal administration⁴⁴, the majority of the available immunogenicity data was from individuals with SC administration (creating a potential mismatch in the effectiveness data and immunogenicity data). Fortunately, immunogenicity data suggests similar antibody titers between the two modes of administration²⁴. An additional difference between immunogenicity and effectiveness studies was the timing of assessment of antibody titers in serum and effectiveness assessment. Whereas immunogenicity was assessed at 4 or 2 weeks after first or second vaccination respectively, the time from vaccination to infection is only reported in one effectiveness study (Wolff Sagy et al.¹⁸, where 3 of 5 infections occur in week 3, and the other two infections occur in week 6).

3. Data extraction: Was the entire process of study screening and data abstraction conducted with double-checks? It has been reported that 17% of data cannot be reproduced in meta-analyses [DOI: 10.1136/bmj-2021-069155]. Therefore, ensuring the accuracy of extracted data is crucial.

We thank the reviewer for their suggestion. Previously, double checking of extracted data had not been completed. We can now confirm that all screening was performed by two authors separately, and data extraction has been conducted by one author and checked by a second author. This is now made explicit in the methods:

Line 425: Screening of the results from each search were conducted independently by two individuals (MTB and SRK). Full details including search terms and inclusion criteria are provided in the supplementary material.

Line 473: Data extracted from tables was conducted by MTB and checked for accuracy by SRK. Where data was extracted from an image, two individuals (MTB, SRK) extracted the data independently with the geometric mean of the two extracted values used and we confirmed that discrepancies between extracted values were always less than 1%.

Some very minor transcription errors were identified by the second author checking the accuracy of the extracted data, and these have now been corrected in the manuscript and all data analysis, with no change to the overall results and conclusions.

4. Statistical analysis:

We are not familiar with hierarchical Bayesian. This part need further check from statisticians.

But why the authors not use the raw reported anitibody? Are they still comparable after the transformation?

We apologise that this was not clear. We believe the reviewer may be referring to the adjustment applied to the data to amend inconsistent reporting of geometric mean titers (GMT) of antibodies between studies – since this is the only adjustment to the raw data. The adjustment made to GMTs to standardise across studies was applied because the GMTs are calculated differently in different studies, and it is possible to precisely standardise these. Specifically, studies used one of two approaches; (a) assigning a value of “1” to the titers below the detection limit, or (b) assigning a value that is half the limit of detection (assign “25” to the titer given the limit of detection (LOD) is 50). In both cases these values were then used to calculate the GMT.

Neither approach is 'better' than the other (in the sense that both use arbitrary values they assign to sero-negative individuals when computing the GMT). However, the choice of using 1 or 25 can create variability between studies. For example, in study Frey 2013, they reported 26% of people were sero-negative. The GMT using method (a) is 29 whilst method (b) is 64. Therefore, whether a value of 1 or 25 was assigned to these seronegative values would significantly affect the estimated GMT from a given study if there are lots of seronegative individuals. Since most studies used the method (a) (ie: assigned a value of 1) we applied the correction (or transformation) to convert GMTs from the other studies that used method (b) to what they would be had method (a) been used. Since converting GMTs between the different methods can be precisely calculated, this adjustment is an obvious means of improving comparability of the studies.

We now provide clarification on this adjustment of the data in the text:

Line 568: When antibody titers were below the limit of detection, this was handled in different ways across the different studies. Some studies had assigned values of "1" to the titers below the detection limit before calculating the GMT, whilst other studies assigned those values as half the limit of detection (assign "25" to the titer given the limit of detection (LOD) is 50). Subsequently, the reported GMTs use different methods for the calculation. For consistency in our analysis, we adjusted the GMTs from the latter studies (those who assign the values below LOD as "25") to reflect the former approach (assign values below the LOD as "1"). Using the reported number of seropositive samples (those above the limit of detection), we can reassign the titers below the LOD from "25" and set those values as "1" with the resulting adjusted GMTs now being consistently calculated across studies.

Results

1. Flowcharts (Figure S1):

a. 67 studies excluded via automation tools. What tool was used? What are the exclusion criteria for that tool, and has any study validated the accuracy of this tool?

In the original submission, this designation was used to indicate studies that were removed because of keyword searches of the titles (performed in endnote) that identified the studies as reviews or animal studies. However, we have removed this partially-automated step and instead reviewed these 67 articles manually (title and abstract screening followed by full text) and indicated the reason for exclusion more precisely in the PRISMA flowchart (Fig S1).

b. Please provide a detailed definition for each criterion. For instance, what is considered a wrong study design, and how is a wrong population defined?

We apologise for a lack of clarity here. As mentioned in previous comments we have updated our systematic search and provided more detail on the process and inclusion criteria. Our goal was to capture as many estimates of vaccine effectiveness as possible, thus we include all studies for which an estimate of vaccine effectiveness was possible. Studies were only excluded where they did not have the required data to calculate vaccine effectiveness – sometimes this was because they did not include estimate of the at-risk population to match the case data among vaccinated and unvaccinated individuals (which is what we had previously meant by “wrong population” – but this was not an ideal categorisation as it was really the lack of appropriate data to calculate the outcome of interest. We have now more rigorously reported why studies could not be included (see PRISMA chart above):

Line 441: Studies were included where they considered populations at-risk of mpox, where the intervention was vaccination with a vaccinia-based vaccine, the control or reference group was unvaccinated people (also at-risk of mpox), and where the outcome was mpox infection/incidence. For inclusion in our analysis, a study needed to include an estimate of VE, or report sufficient data such that VE could be estimated. This required data for:

- *Cases of mpox in vaccinated and unvaccinated cohorts or populations along with an estimate for the population at risk for both unvaccinated and vaccinated groups (for cohorts, case-coverage and secondary contact study types); or*
- *Vaccination status of individuals with cases of mpox, and a control group who were uninfected and for whom vaccination status was reported (for case-control studies).*

Studies were excluded when vaccinia-vaccination was used as a post-exposure prophylaxis intervention rather than as a pre-exposure intervention.

And

Supplementary Methods: In order to calculate vaccine effectiveness from a given study, and thus include the study in our analysis, we required the following data:

- *In the case of studies based on population surveillance data (case-coverage), the incidence of infection must be provided in both vaccinated and unvaccinated at-risk groups.*
- *In the case of secondary-contact studies, the total number of secondary-contacts/exposed individuals (disaggregated by vaccine status) must be provided, along with the number of secondary-cases in vaccinated and unvaccinated individuals.*
- *In the case of cohort studies, the cohort size of both vaccinated and unvaccinated groups must be provided along with the number of infections in each group.*
- *For case-control studies we require a break-down of infection data by vaccination status along with an uninfected control group split by vaccination status as a comparator.*

2. Lines 116-120: Was no vaccination the reference group when estimating effectiveness for all groups? What is the average follow-up duration for each vaccine? Due to waning of efficacy, it is important to describe the duration when comparing effectiveness across different vaccines.

We thank the reviewer for identifying the lack of clarity here. Yes, no vaccination was always the reference group when estimating vaccine effectiveness, we now explain this in the methods:

Line 501: Unvaccinated individuals are used as the reference group for calculating vaccine effectiveness. When comparing relative effectiveness between vaccines we compare the Odds ratio between the two vaccines (i.e. $OR_{1,2} = OR_1/OR_2$).

We have now added the follow-up duration to sup. table 1 where appropriate. For test-negative designs there is no follow-up but there is a range of times since vaccination – which were not reported, we now highlight this in table S1. We have added additional information surrounding the follow-up time into the results:

Line 102: Information on the timing of cases and follow-up following vaccination was usually not reported (Table S1). For first generation vaccines the observation periods started after routine vaccination ceased in the area⁹ and subsequently infections occurred long after vaccination. The case-control and case-cohort studies of MVA-BN monitored infections during a period of ongoing vaccination. Only individuals who were vaccinated more than 14 days ago, were considered vaccinated within these studies, however the time between vaccination and infection was not reported. In the single cohort study, participants had 21 weeks follow up time with infections in vaccinated individuals occurring at 3 weeks and 5 weeks post-vaccination¹⁸.

Effectiveness Study	Vaccine	Country	Study Design	Observation period	Follow-up *	Population/ Control	Total cases	Disaggregation#	Reported Doses	Included in Data Analysis ^A
Breman 1980 ¹	First Generation	DRC + Nigeria	Secondary Contacts	1970-1979	Not reported	447	4	Spatial	At least 1	Included
Fine 1988 ²	First Generation	DRC	Secondary Contacts	1980-84	Not reported	834	36	Spatial	At least 1	Not included. Major overlap with (Jezek 1988)
Jezek 1986 ³	First Generation	DRC	Secondary Contacts	1980-1984	Not reported	2510	56	Spatial	At least 1	Not included. Major overlap with (Jezek 1988)
Jezek 1988 ⁴	First Generation	DRC	Secondary Contacts	1981-1986	Not reported	2278	69	Spatial and age	At least 1	Included
Rimoin 2010 ⁵	First Generation	DRC	Case-Coverage	2005-2007	Not reported	Estimated### #=#**	760	Age	At least 1	Included
Nolen 2015 ⁶	First Generation	DRC	Secondary contacts	2013 (Jul-Dec)	Not reported	97	44	None	At least 1	Included
Whitehouse 2021 ⁷	First Generation	DRC	Case-Coverage	2011-2015	Not reported	Estimated	1057	None	At least 1	Included
Wolff Sagy 2023 ⁸	MVA-BN	Israel	Cohort Study	2022 (Aug-Nov)	90-147 days	2054	18	Weekly	At least 1	Included
Payne 2022a ⁹	MVA-BN	US	Case-Coverage	2022 (Jul-Sep)	Not reported	Estimated	5402	Weekly	At least 1	Not included. Major overlap with (Payne 2022b)
Payne 2022b ¹⁰	MVA-BN	US	Case-Coverage	2022 (Jul-Oct)	Not reported	Estimated	9544	Weekly	1 or 2	Included
Bertran 2023 ¹¹	MVA-BN	UK	Case-Coverage	2022 (Jul-Oct)	Not reported	89240	460	Weekly	At least 1	Included
Dalton** 2023 ¹²	MVA-BN	US	Case-Control	2022-2023 (Aug-Mar)	Not reported	608	309	None	1 or 2	Included
Deputy** 2023 ¹³	MVA-BN	US	Case-Control	2022 (Aug-Nov)	Not reported	8649	2266	None	1 or 2	Included
Rosenberg** 2023 ¹⁴	MVA-BN	US	Case-Control	2022 (Jul-Oct)	Not reported	507	252	None	1 or 2	Included

3. Lines 145-146: Why did the author choose to use GMT 4 weeks after 1 dose MVA-BN and GMT 2 weeks after 2 dose MVA-BN ? When was the antibody titer for 1st Gen detected?

The timing of GMT measurement after vaccination was determined by the data available in the studies. By far the most common sampling meant that titers after the first doses were 4 weeks post-vaccination and the titers after the second dose were 2 weeks post-second-dose. This was likely due to when investigators expected peak titers to be achieved after a primary and secondary vaccination. This is now explained in the methods:

Line 144: To investigate the immunogenicity of different vaccination strategies, we aggregated data on the geometric mean vaccinia-binding titers (GMT) reported 4 weeks after one dose of MVA-BN and 2 weeks after two doses of MVA-BN. This is because these were the most common times sampled after first and second doses of MVA-BN vaccination across all studies (Table S2). This also coincides with when the peak titers were observed after each subsequent dose²³⁻²⁵.

Unlike the MVA-BN data, both the data for titers and vaccine effectiveness after 1st Gen vaccination were collected many years after 1st Gen vaccination. That is, 1st Gen vaccination was always historic vaccination in our analysis. Antibody titers tend to plateau and remain steady (or decay with a very long half-life) after historic vaccination (Amanna et al.). Thus, we assume that the immunogenicity studies of historic vaccination are comparable to those in the VE studies of 1st Gen vaccines. By using these baseline data in the historically vaccinated cohorts we limit the confounding due to variation in assay design as the assays used across the studies are comparable. This is explained in the methods:

Line 149: For first generation vaccines, we use the baseline immunogenicity data (prior to receiving MVA-BN vaccine) for groups who had evidence of a previous smallpox vaccination. This is assumed to reflect the long-term vaccinia-binding titers maintained by individuals after receiving a first generation vaccine many years earlier.

And limitations addressed in the discussion

Line 363: A mismatch also exists in comparing historic first-generation vaccination. The effectiveness data are from studies in the Democratic Republic of Congo in the 1980s to 2010s, who were vaccinated prior to 1980⁹, whereas the immunogenicity data are from individuals in the United States (who in most cases were vaccinated >40 years earlier). As well as differences in the viral clades between these outbreaks^{43,44}, timing of previous vaccination in the historic vaccine effectiveness studies is not recorded and may not be well matched to the immunogenicity studies. Evidence of a very slow long-term decay of antibodies^{37,45-47} suggests that time-since-vaccination may not be critical in comparing these groups many years after vaccination. However, it was not possible to match for age of vaccination, health status, or other demographic variables and their effects on immunogenicity and protection are unknown. Previous work has shown that vaccination in childhood confers longer protection than vaccination in adulthood⁴⁸. Further investigation of the risk of breakthrough infection in historically smallpox vaccinated cohorts are required to confirm this assumption and improve our understanding of the duration of mpox immunity from vaccinia vaccines⁴⁹.

4. Figure 4: Most points were between 0 to 0.5 years, very few points were in 1 to 2 years. Is this model reliable? Why is confidence interval so narrow?

We agree that additional data points would help strengthen the model but we were limited to the available data. However, importantly, our model incorporates the observations across many studies and across these studies there are a total of several thousand individuals (9936 total enrolled in the trials - although some arms of these studies were not relevant to our analysis and excluded). The relatively narrow credible intervals on the biphasic model, capture the uncertainty in the GMT (not individual titers), but individual titers will be distributed around these GMT values. Because the GMTs are calculated from many individuals in each study, the GMTs are fortunately robust compared to the distribution of actual individual titers, and thus the CIs of this GMT are relatively narrow. Further, the narrow CIs likely also indicate consistency in the trends of decay observed across different studies.

Discussion

1. Lines 322-333: The author recommended a longer interval for booster vaccination based on the humoral immunity evidence presented in this study. However, considering the significance of cellular immunity in formulating vaccination strategies, could the author provide any supporting evidence for cellular immunity?

We thank the reviewer for this suggestion. Unfortunately, T cell responses were reported in only 3/13 immunogenicity studies). Therefore, it was not possible to ascertain whether T cell responses might also be predictive of protection. Fortunately, animal studies and natural history studies suggest a primary role for antibodies (and this has been used for immunobridging by regulators). However, we have added a comment on the potential role for cellular immunity.

Line 379: The evidence from animal studies showing a role for antibodies in protection from mpox¹⁴, and the role of antibodies as a correlate of protection for smallpox¹³, prompted us to consider antibodies as a correlate of protection in this study. However, despite finding an association between vaccinia-binding and protection, this is likely not an optimal correlate against mpox. In part this is because, even though vaccinia-binding titers are correlated with in vitro neutralizing antibody titers to mpox after primary vaccinia-vaccination²², cross-recognition between vaccinia and monkeypox virus will likely be inconsistent when exposure histories vary (e.g. when an individual experiences a primary exposure to an mpox antigen rather than vaccinia²²) or against different viral variants. Our ability to study neutralizing antibodies and other potential correlates of immunity was limited by the available data - with much less data on neutralizing antibodies than binding titers, and very limited reports and a lack of standardized assays for measuring cellular immunity. Further work is necessary to compare different measures of immunogenicity and define optimal correlates of protection for mpox.

2. Lines 361-391: The author mentioned significant differences among certain paired studies involving efficacy and immunity. Could the author include a table delineating the characteristics of these paired studies for readers to easily comprehend the distinctions? Additionally, has the author attempted to select studies that are seemingly comparable for sensitivity analysis to evaluate the robustness of the results?

We thank the reviewer for highlighting the lack of clarity on this point, and along with the response to methodology 2b above, we have now included a supplementary figure making it clear how immunogenicity studies and vaccine effectiveness studies were linked (figure S5). Importantly, the link between immunogenicity and effectiveness studies is not a one-to-one pairing but a many-to-many grouping. This is because sufficient demographic links between the populations in the VE and immunogenicity studies were not possible to establish. What details were available regarding the populations in each study are provided in Tables S1 and S2. It is clear from this that there are demographic differences between these studies, and thus we discuss this limitation in the text:

Line 347: Another limitation is that the VE data and immunogenicity data came from independent cohorts and studies. Thus, there is no guarantee that the populations are well matched. Specifically, there is a significant mismatch in the demographics of the vaccine effectiveness studies and the immunogenicity trials. For example, the population considered in the US study for VE¹⁷ considered only men (sex assigned at birth or gender identity) aged between 18-49. On the other hand, the clinical trials of

antibody responses post vaccination were all tested on populations of both men and women and featured slightly different age ranges. Further, even though the majority (85.7%) of individuals in the US received their second dose of MVA-BN via intradermal administration⁴², the majority of the available immunogenicity data was from individuals with SC administration (creating a potential mismatch in the effectiveness data and immunogenicity data). Fortunately, immunogenicity data suggests similar antibody titers between the two modes of administration²⁴.

Reviewer #2 (Remarks to the Author):

This paper by Berry et al uses a similar technique used by the same group for COVID-19 vaccines to link vaccinia binding antibody toters with vaccine effectiveness. The results are that antibodies correlate with efficacy according to a logarithmic - linear distribution and that vaccine efficacy can be projected to be reasonably high after 2 doses for many years based on antibody decay kinetics.

The work is of high significance to the field because vaccination strategies to protect against subsequent MPox outbreaks are needed and optimization of protection amongst high risk groups is required.

The work represents a quandary of sorts. The analysis contains no major flaws and probably approximates the best analysis that can be done with available data to optimize Mpx vaccine allocation now and in future outbreaks. That said, the available data is much less useful than the data this group used to conclude that neutralizing antibodies are a good surrogate of protection against SARS-CoV-2. The authors do a beautiful job of outlining these limitations in the discussion which include the fact that the analysis only contains 3 types of vaccine studies (which is very low to form a meaningful correlation as in Fig 3), that neutralizing Abs are often not measured in most of these studies, that the studies include a mix of case-control & cohorts studies in different populations which are not well matched, that vaccine formulations vary over time, that behavioral confounders likely vary between study populations, and that other potentially important Mpx immune responses go unmeasured, For these reasons, the study does not provide strong enough evidence to establish antibodies as a formal correlate and the authors acknowledge this.

Overall, the conclusions are only somewhat supported by the analysis despite an adequate and thorough methodological approach. Despite the level of evidence being only low-moderate, the results are probably actionable because this is the strongest type of analysis available for the problem to my knowledge, and the problem is rather urgent.

We thank the reviewer for their supportive comments of the work. We agree with the reviewer that there is a considerable list of limitations for this analysis, and are glad that the reviewer feels this has been transparently reported. We agree with the reviewer's sentiments that the data is not ideal, and yet public health decision making is still required and thus we feel that our integration of the available data serves an important need in a timely manner.

I have some minor suggestions:

1) Show the strength of binding / neutralizing antibody correlation in past studies and describe in a. bit more detail how these studies were done.

We thank the reviewer for this suggestion. We have now added the correlation coefficient and described the result in more detail:

Line 112: We focused on vaccinia-binding titers because Zaeck et al have shown that endpoint antibody binding titers correlate well with neutralizing antibody titers against monkeypox virus, using samples from recently vaccinia-vaccinated individuals (reported $r = 0.82$, $p < 0.0001$)²². Moreover, endpoint vaccinia-binding titers are the most commonly reported measure of immunogenicity allowing comparison between multiple studies.

2) When describing VE, state over what time interval. Was this interval even the same in all listed studies?

We thank the reviewer for this comment. Unfortunately, little to no information has been provided regarding the time intervals of the VE studies. Even data on the median time since vaccination has not been provided in most studies. We now add any information provided by the authors on the time interval for VE assessment to Table S1.

(Also, see response to reviewer 1, comment Results 2)

3) Figure 1B is too squished and would be easier to read with a wider x-axis.

We thank the reviewer for this suggestion. We have now spaced-out Figure 1B and included an x-axis tick to improve readability.

4) The horizontal lines in Fig 2 are hard to see. I suggest reformatting

We thank the reviewer for this suggestion. We have made this figure clearer by removing the horizontal dashed lines, which previously indicated the credible intervals – instead we now use a shaded region to indicate the credible intervals and solid lines for the means.

5) Fig 3: "The contours represent the lines of equal probability of the normalized joint..... etc..." I do not know what this means. This should be stated with less jargon.

We thank the reviewer for raising this lack of clarity. We have now made this more explicit with more explanation:

Figure 3: Relationship between vaccine effectiveness and the vaccinia-binding GMT. The contour lines, represent the 20%, 40%, 60% and 80% highest density regions of the joint-posterior distribution (i.e. smallest areas that contain x% of the posterior samples) for the different vaccines. The association between antibody titers and effectiveness (solid black line) is fitted using all of the underlying data (accounting for the interstudy heterogeneity using a hierarchical model structure) (Table S5). The solid black line indicates the best estimate (median of posterior), and shaded region show the 95% credible intervals of the predicted effectiveness at different GMTs.

6) For bi-phasic decay, how was the timing of the inflection point selected?

The biphasic decay model used here is a double exponential model, which doesn't require us to define a fixed inflection point. Instead, the model captures two populations, long and short lived antibodies, that decay at different rates, generating a quick initial decay and a

slow long-term decay, and all parameters in the model as estimated by model fitting rather than selected by the authors. The inflection point then arises automatically from the estimated proportion of long and short lived antibodies and the decay rates of each, which are all estimated from model fitting and not specified in advance. The model formulation and parameter estimates are provided in the main text and supplementary material:

Line 580: We fit a model of biphasic exponential decay to the vaccinia-binding antibody titers. This model has two compartments, long (x_l) and short (x_s) lived antibody titers, which each are assumed to decay with rates δ_l and δ_s , respectively. Thus, the total antibody titer at time, t , is given by $x(t) = x_l(t) + x_s(t)$, where,

$$x_s(t) = x_0 f e^{-\delta_s t},$$

$$x_l(t) = x_0 (1 - f) e^{-\delta_l t},$$

and where x_0 is the antibody titer at $t = 0$ (which is defined as a maximum of 14 days after final dose or 28 days after the first dose), and f is the fraction of the initial antibody titer that is short-lived.

And Table S6 provides the parameter estimates of the fitted decay model.

Parameter	Dosing Scheme	Symbol	Prior	Estimate (95% Credible Interval)
Initial GMT	1-Dose	x_0	$\log(x_0) \sim N(0,10)$	80 (63-102)
	2-Dose (28 days)			673 (537-854)
	2-Dose (730 days)			2136 (1596 -2864)
	3-Dose			2211 (1685)
Slow-decay rate (long-lived antibodies)		δ_l	$\delta_l \sim N(0,1)$	0.0028 (0.0007- 0.0050)
Fast-decay rate (short-lived antibodies)		δ_s	$\delta_s \sim N(0,1)$	0.23 (0.20-0.27)
Proportion of fast decaying antibodies	1-Dose	f	$f \sim U(0,1)$	0.89 (0.87-0.92)
	2-Dose (28 days)			0.94 (0.93-0.95)
	2-Dose (730 days)			0.74 (0.65-0.81)
	3-Dose			0.70 (0.61-0.77)

7) I feel as if the optimization of timing of dose #2 is important and should be mentioned in the abstract

We thank the reviewer and have now added reference to these results in the abstract:

Line 38: We find that delaying the second dose of MVA-BN vaccination will provide more durable protection and may be optimal in an outbreak with limited vaccine stock.

Reviewer #3 (Remarks to the Author):

The authors performed a systematic search and meta-analysis of immunological and observational studies for Mpxv vaccines, to establish the correlation of protection (CoP) between an immune marker (vaccinia-binding ELISA endpoint titer) and vaccine effectiveness. The authors then estimated the antibody kinetics using aggregated data collected two years after the vaccinations, and predicting the long-term vaccine effectiveness against Mpxv by extrapolating the antibody kinetics to 10 years with the previously estimated

CoP. The study largely inherited methods used for several COVID-19 studies published by these authors.

My main concern is that one of the main claims of this study was supported by extrapolating the established CoP of vaccinia-binding ELISA endpoint titer and vaccine effectiveness from 2-year data to a longer time span up to 10 years. Although the authors used a separate antibody kinetic model to predict the antibody titer level at ten years, they used a constant CoP that was estimated using data collected relatively shorter after vaccination. The key question will be whether or not the CoP identified by the authors will be a constant over time. If not, then the extrapolated vaccine effectiveness may be biased because the CoP was completely different from the early after vaccination. Such issue has been seen in COVID-19 where ELISA and even neutralising antibody derived from inactivated vaccines against Omicron were barely detectable, but actually the vaccine effectiveness against fatal outcomes were still >90%. Therefore, the current study oversimplifies this question with extrapolations using only a constant CoP, which will weaken their findings and claims.

We thank the reviewer for highlighting an additional limiting assumption in our extrapolation on long-term vaccine effectiveness. This limitation is important, and we now discuss this in more detail in the discussion. We note that the correlate of protection we derive includes a mixture of effectiveness trials shortly after vaccination and those in people with historic first-generation vaccine effectiveness. It is an assumption that these are governed by the same correlate of protection – however, encouragingly, these effectiveness studies on VE long after vaccination show high efficacy – and comparable to that achieved shortly after vaccination (which were induced by similar titres (i.e. orange and blue in figure 3 are very similar). These important points are now discussed in the discussion.

Line 393: Finally, it is not clear that a correlate of protection identified shortly after MVA-BN vaccination will continue to predict VE over time as immunity wanes. However, encouragingly, the data used in this analysis on the VE and antibody titers from first generation vaccinia-vaccination are all studying individuals long after vaccination and reveal similar titers and VE to a single dose of MVA-BN (Fig. 3), consistent with the possibility that vaccinia-binding continues to predict VE long after vaccination.

As the authors included the long-term vaccine effectiveness predictions as a part of the study, it is important to clarify at what time the antibody and vaccine effectiveness were measured throughout the manuscript. Currently, this information was mostly unclear when reading the first four sections of results. For example, when the authors were describing estimating the correlation between antibody and vaccine effectiveness for the same vaccine type/dose, it is unclear whether the time points were also matched for these estimates.

We have now added detailed information on the timing of studies for VE where available in the supplementary Table S1. However, importantly, most VE studies did not provide any information on individuals' time since vaccination. Further, we have now made explicit throughout the results/discussion what data was available on timing of vaccination, e.g.:

Line 144: To investigate the immunogenicity of different vaccination strategies, we aggregated data on the geometric mean vaccinia-binding titers (GMT) reported 4 weeks after one dose of MVA-BN and 2 weeks after two doses of MVA-BN. This is

because these were the most common times sampled after first and second doses of MVA-BN vaccination across all studies (Table S2). This also coincides with when the peak titers were observed after each subsequent dose²³⁻²⁵. For first generation vaccines, we use the baseline immunogenicity data (prior to receiving MVA-BN vaccine) for groups who had evidence of a previous smallpox vaccination. This is assumed to reflect the long-term vaccinia-binding titers maintained by individuals after receiving a first generation vaccine many years earlier.

And

Line 179: The VE studies on MVA-BN were conducted during a period of ongoing vaccination and in most cases did not report the time between vaccination and infection (Table S1). Subsequently, we match this effectiveness data to the peak vaccinia-binding titers, which occur shortly after vaccination. First-generation vaccines were administered in the DRC prior to 1980⁹ (when routine vaccination officially ceased), whilst the effectiveness studies range between 1970 and 2015. Subsequently in some studies vaccination occurred more than 35 years prior to infection. In order to account for these effectiveness studies being in individuals many years post-vaccination, we match this effectiveness data with the immunogenicity data for historically vaccinated cohorts (i.e. individuals vaccinated years earlier, and enrolled in an MVA-BN vaccine trial, but we use their baseline vaccinia-binding titers from these individuals, before they receive the MVA-BN vaccine).

And

Line 358: An additional difference between immunogenicity and effectiveness studies was the timing of assessment of antibody titers in serum and effectiveness assessment. Whereas immunogenicity was assessed at 4 or 2 weeks after first or second vaccination respectively, the time from vaccination to infection is only reported in one effectiveness study (Wolff Sagy et al.¹⁸, where 3 of 5 infections occur in week 3, and the other two infections occur in week 6). A mismatch also exists in comparing historic first-generation vaccination. The effectiveness data are from studies in the Democratic Republic of Congo in the 1980s to 2010s, who were vaccinated prior to 1980⁹, whereas the immunogenicity data are from individuals in the United States (who in most cases were vaccinated >40 years earlier). As well as differences in the viral clades between these outbreaks^{43,44}, timing of previous vaccination in the historic vaccine effectiveness studies is not recorded and may not be well matched to the immunogenicity studies.

Figure 1 - Suggest the authors to demonstrate the two panels as forest plots, where both individual and pooled estimates for subgroups can be clearly demonstrate. Also, it would be helpful to also report the I statistics for each subgroup to assess the heterogeneity of individual estimates within each subgroups.

We thank the reviewer for this comment. Since so much of the focus of the analysis is on comparison of the results between vaccine regimens, we opted to present the data in a manner we feel provides the most direct cross-regimen comparison. However, we agree with the value of a forest plot for assessing the overall estimates and the between study heterogeneity. We have now included a forest plot of the results from Figure 1 in supplementary Figure S3.

Regarding measures of between study heterogeneity, an I^2 statistic is not typically reported for meta-analysis in a Bayesian framework. Instead, we quantify the heterogeneity between studies with the between study variability parameter (Tau2) with the sigma parameters displayed in the tables of parameters in the appendix (Table S3-S6).

Vaccine effectiveness were all extracted from observational studies, which unavoidably faced various types of confounders. Especially, there were several studies were performed in close contacts. Therefore, it is important to address this limitation and discuss the potential impacts of measurements from these study designs.

We thank the reviewer for highlighting these confounders. In our discussion we consider the different study designs and highlight some of the aspects that act as confounders. We have now added explicit discussion on the types of confounders that can arise through secondary contacts along with our discussion of the confounders that arise due to other study types.

Line 322: A major challenge in using non-randomized studies is appropriate matching of control groups (in case-control studies) and identification of the at-risk population (in case-cohort studies). Differences in matching cases to controls significantly affect the reported levels of protection^{15,16,19}. In addition, large case-cohort studies attribute the reduction in case numbers to vaccination, but this may not control for confounders such as differences in behavior (which have been associated with a reduction in transmission in Italy prior to the commencement of vaccination⁴¹). Analysis of secondary contacts cannot always account for the number and significance of interactions with the contacts and thus can introduce unmeasured confounding. Further, in these studies conducted in the DRC, the status of individuals as vaccinated or unvaccinated was typically the result of the timing of a mass vaccination program that ceased in 1980⁹ and so was heavily skewed by age. These

confounders may contribute to the substantial heterogeneity in VE observed across studies (Fig. 1). We can partially account for unmeasured confounding by using a hierarchical model to account for inter-study variability, but systematic biases that result from unmeasured confounding are unable to be completely excluded.

REVIEWERS' COMMENTS

Reviewer #1 (Remarks to the Author):

All my concerns have been addressed.

Reviewer #2 (Remarks to the Author):

The authors have provided thorough responses to comments from all 3 reviewers. I have no further suggestions.

Reviewer #3 (Remarks to the Author):

The authors have addressed all my concerns.